# Assessing Aflatoxin Exposure Risk from Peanuts and Peanut Products Imported to Taiwan

**DOI:** 10.3390/toxins11020080

**Published:** 2019-02-01

**Authors:** Keng-Wen Lien, Xin Wang, Min-Hsiung Pan, Min-Pei Ling

**Affiliations:** 1Institute of Food Science and Technology, National Taiwan University, Taipei City 11031, Taiwan; d98b47102@ntu.edu.tw; 2Food and Drug Administration, Ministry of Health and Welfare, Taipei City 11561, Taiwan; 3Department of Food Science, National Taiwan Ocean University, Keelung City 202, Taiwan; wangzjxxin@gmail.com; 4Institute of Food Science and Technology, National Taiwan University, Taipei City 11031, Taiwan; mhpan@ntu.edu.tw

**Keywords:** aflatoxin, risk assessment, peanut, imported food

## Abstract

Aflatoxins are highly toxic and cause disease in livestock and humans. In order to assess Taiwan population exposure to aflatoxin from peanuts and peanut products, a total of 1089 samples of peanut candy, peanut butter, and peanuts etc. were collected in the period from 2011 to 2017 and analyzed using a liquid chromatography/tandem mass spectrometer. The overall mean contamination levels of aflatoxin in peanuts and peanut products were 2.40 μg/kg of aflatoxin B1, 0.41 μg/kg of aflatoxin B2, 0.19 μg/kg of aflatoxin G1, and 0.03 μg/kg of aflatoxin G2. We use margin of exposure (MOE) as a tool to improve food safety management. According to MOE levels of aflatoxins in peanuts and peanut products from China, Indonesia, Thailand, the United States, and the Philippines were above the safe lower limit of 10,000, indicating an absence of public health or safety risk for the majority of the population. However, products from Vietnam were under the MOE safe lower limit, suggesting that regulatory actions must be continued to avoid excessive consumer exposure.

## 1. Introduction

Aflatoxins are toxic metabolites produced by Aspergillus, which includes the species *A. flavus*, *A. parasiticus*, and *A. nomius* [1]. The types of Aflatoxins presenting most significant public-health concerns are aflatoxin B1, aflatoxin B2, aflatoxin G1, aflatoxin G2, aflatoxin M1, and aflatoxin M2, all of which are highly toxic and contain carcinogenic compounds that cause disease in livestock and humans [2,3]. The International Agency for Research on Cancer (IARC) has classified aflatoxin B1 as a Group 1 human carcinogen [4].

Aflatoxins are found in many tropical and subtropical countries where temperature and humidity conditions are optimal for mold growth and for production of these toxins [5]. Aflatoxins are small, stable molecules that cannot be destroyed by heat treatment or during processing [6,7]. Removal, reduction or inactivation of aflatoxins in food and feedstuffs is a major global concern. In Taiwan, people of all age groups widely consume peanut and its products. Taiwanese make desserts with peanut flour and cook many dishes with roasted peanuts, like peanut candy, spicy garlic, and other Chinese dishes. Taiwan produces its own peanuts, but due to the impact of Taiwan’s climate, a shortage of arable land, and high personnel costs, which result in small peanut harvests and high prices, nearly 70% of its peanuts and its product are imported from abroad. Thus, monitoring and prevention of aflatoxins in peanut foods is an important issue.

Previous studies have demonstrated high incidences of aflatoxins contamination in nuts, a product whose consumption has increased in Taiwan over the last decade [8]. The Taiwan Food and Drug Administration (TFDA) Imported Food Control Plan was designed to protect consumer food safety by investigating aflatoxins levels in imported peanuts and peanut products. The aim of this study is to derive dietary exposure estimates for aflatoxin in peanuts and peanut products imported to Taiwan to provide health authorities with a risk assessment of aflatoxin contamination in imported foods and a reference for imported food management.

## 2. Results and Discussion

### 2.1. Aflatoxin Contamination Concentrations and Daily Consumption Data

We assessed the exposure of population groups of various ages to aflatoxins in imported peanuts and peanut products, and the resultant health risks. A total of 1089 samples of peanut candy, peanut butter, and peanut products imported from various countries were collected in the period from 2011 to 2017. All samples were analyzed using liquid chromatography/tandem mass spectrometer. This method of analysis is considered reliable and was carried out in accordance with the TFDA Official Method [9]. The limit of quantification was 0.2 μg/kg, 0.1 μg/kg, 0.2 μg/kg, and 0.1 μg/kg for aflatoxin B1, B2, G1, and G2, respectively.

As shown in Table 1, of the 1089 peanut samples taken, aflatoxins (B1 + B2 + G1 + G2) were found in 25%, 24% tested positive for aflatoxin B1, 17% for aflatoxin B2, 3.3% contained aflatoxin G1 and 1.8% tested positive for aflatoxin G2. Of the peanuts and peanut products sampled, 3.1% exceeded the maximum levels for aflatoxins. The highest recorded level was 432.0 μg/kg of aflatoxin B1 in a sample from Indonesia. The overall mean contamination levels of peanuts and peanut products were 2.40 μg/kg of aflatoxin B1, 0.41 μg/kg of aflatoxin B2, 0.19 μg/kg of aflatoxin G1, and 0.03 μg/kg of aflatoxin G2.

Table 2 concentrations are from a monitoring study from those presented in Table 1. The percentage of positive samples varied greatly between countries (13–47%) as shown in Table 2. In our survey, there are five samples (1 roasted peanuts with coconut juice, 2 peanut butter products, and 2 peanuts samples) were contaminated with high levels of aflatoxins, and their total aflatoxins concentrations (the sum of B1, B2, G1 and G2) were higher than Taiwan’s regulatory limit (15 ppb). These products were either returned or destroyed. The average aflatoxins contamination levels in the products from the countries studied were 0.947 ppb for China, 4.180 ppb for Vietnam, 11.805 ppb for Indonesia, 1.562 ppb for Thailand, 1.372 ppb for the United States, 1.053 for the Philippines, and 1.784 ppb for others. The highest levels of aflatoxins found in products from China, Vietnam, Indonesia, Thailand, the United States, the Philippines, and others (Japan) were 8.1, 258.3, 412.0, 143.0, 28.0, 14.2, and 441.0 ppb, respectively. Aflatoxins were easily detected in peanuts and peanut products (peanut candy, peanut butter, peanut flour, etc.). Raw peanuts are milled to make processed peanut products such as peanut candy, peanut ice cream roll, and peanut powder. Milling, cutting or grinding exposes peanuts to oxygen and mold more readily because the surface area is increased. This may be why the frequency of aflatoxins contamination is higher in processed peanut products than in raw peanuts in their shells [8].

In Taiwan, border inspection techniques include visual inspection, hygiene-quality examination, and document verification. From 2011 to 2017, the top six countries of origin for peanuts and peanut products by weight were Vietnam (42.07%), China (33.93%), the United States (4.10%), Thailand (1.09%), Philippines (0.59%), and Indonesia (0.53%) shown in Figure 1. Because the food mass of imported peanuts and peanut products consumed (kg/year) from these countries differs, the import-to-production factors of this study were included. For example, if 10% of peanuts and peanut product are imported, an import-to-production factor of 0.1 is derived.

Another post-market survey of 1827 Taiwan commercial peanut products was conducted from 1997 to 2011. The average percentage of positive samples was: 32.8% for peanut candy, 52.8% for peanut butter, 7.8% for peanuts, and 44.1% for peanut flour. Aflatoxins were detected in 32.7% of samples and the levels from 0.2 μg/kg to 513.4 μg/kg [8]. In our survey, the contaminated percentage of sampled peanuts and peanut products in 2011 to 2017 was 15%, 22%, 24%, 19%, 29%, 38% and 19%, respectively. Further, the percentage of positive samples varied greatly between countries (13% to 47%) (See Table 2).

The estimated mean exposure to the sum of the highest aflatoxins exposure was 0.0145 ng/kg bw/day in 1–2 year olds for the whole group, and the consumer only group was 0.458 ng/kg bw/day in 1–2 year olds. Chen et al. summarized survey data from 1997 to 2011 that concerned aflatoxin contamination in peanuts and peanut products in Taiwan [8]. Their results show the probable mean daily intake (PDIM) of aflatoxins from peanut products in Taiwan was estimated as 1.07 ng/kg and 0.92 ng/kg of body weight per day for males and females, respectively. In the Second French Total Diet Study, the mean exposure to the sum of the aflatoxins was 0.0019 ng/kg bw/day in adults and 0.0013 ng/kg bw/day in children (lower bound). For the upper bound, mean exposure was 0.89 ng/kg bw/day in adults and 1.56 ng/kg bw/day in children [10].

Overall, it is difficult to compare aflatoxins exposure results from different countries because of differences in analytical limits, which may have resulted in high or low aflatoxins censoring. Furthermore, differences in sampling methods make it inappropriate to compare mean contamination concentrations between the food groups.

### 2.2. Risk Characterization

There was no safe level established for aflatoxins exposure due to its genotoxic carcinogenic potential [11]. Therefore, the risk was characterized using the margin of exposure (MOE) approach. MOE is calculated by measuring the ratio between the dose that produces a specified effect (benchmark dose, BMD) and the estimated human aflatoxins intake. In this study, MOE value is 170 ng/kg bw per day, which was estimated by the European Food Safety Authority (EFSA) based on carcinogenicity data in rats exposed to aflatoxins [11].

In Table 3, MOE was used to consider possible safety concerns arising from the presence of peanuts and peanut products imported from different countries. By definition, a larger human risk occurs when MOE is <10,000 [11]. Even in the consumer only scenario, the MOE of aflatoxins in products from Indonesia, Thailand and the Philippines was above this safe lower limit. Risk assessors at the EFSA use the MOE approach to consider possible safety concerns arising from the presence of genotoxic and carcinogenic substances in food or feed. When the calculated MOE is ≤10,000, the EFSA estimates the existing carcinogenic risk as low and suggests that these substances be treated with low priority [12].

Figure 2 shows the estimated exposure (whole group) of Taiwanese people (all ages) to aflatoxins in peanuts and peanut products from various countries. The aflatoxins MOE levels in peanuts and peanut products from China, Indonesia, Thailand, the United States, and the Philippines were above the safe lower limit of 10,000, indicating that for the majority of the population, there is no public health or safety risk posed by the consumption of imported peanuts and peanut products. However, the results indicate that consuming products imported from Vietnam may pose potential health concerns. This suggests that regulatory actions must be continued, which includes educating consumers about processed foods with relatively high contamination to avoid excessive exposure.

### 2.3. Risk Management Advice

We suggest that Taiwan’s government needs to continue efforts to reduce dietary exposure of aflatoxins. On the basis of our risk assessment results, the Taiwan government should control dietary exposure to aflatoxins. If the maximum levels of total aflatoxins were changed from 15 to 4 ppb (as in the EU), this might help reduce aflatoxins exposure risk (Figure 3) and raise the MOE of aflatoxins above the safe lower limit. Therefore, the risk to aflatoxins exposure will not constitute a public health problem.

The most effective way of controlling aflatoxins is to monitor food and its raw ingredients for aflatoxins. Strengthening border control, carrying out strict periodic monitoring, and imposing fines on noncompliant products could help reduce aflatoxins production, accumulation and thus contamination in susceptible commodities to an acceptable level, while preventing significant health and economic losses. The results of this assessment may be useful to the TFDA to improve supervision of product improvement by peanut product manufacturers, ensure border inspections are carried out, and realize effective imported food control. Aflatoxins in peanut foodstuffs can also be controlled by improving production practices and maintaining appropriate storage conditions. Food companies should be required by relevant government organizations to test received peanuts for aflatoxins. The potential health risks of aflatoxins may be reduced by educating and enhancing the awareness of farmers and consumers.

## 3. Conclusions

In Taiwan, nearly 70% of peanuts and peanut products are imported from abroad. These peanut food products, as high-risk imported food products, should be actively monitored. If the sanitation standards of aflatoxin levels in Taiwan changed from 15 to 4 ppb, this might reduce consumers’ exposure to aflatoxin contamination from imported peanut products.

## 4. Materials and Methods

### 4.1. Imported Food Product Inspections

Imported food product inspections are conducted on the basis of the “Guidelines for Imported Foods Inspection,” and the “Regulations for the Inspection of Imported Foods and Related Products.” The inspections are based on the provisions of the Act Governing Food Safety and Sanitation, the food sanitation standards for various food products, and related regulations [13,14,15]. In general, the inspection is performed based on a 2–10% inspection rate. According to “Regulations for the Inspection of Imported Foods and Related Products”, Reinforced randomly-selected batch inspections (20–50% inspection rate) or Batch-by-batch inspection (100% inspection rate) may be implemented for the management and control of foods with higher risk of safety. A total of 1089 samples of peanut candy, peanut butter, and groundnuts were collected during imported food product inspections from 2011 to 2017.

### 4.2. Analysis of Aflatoxins

Analysis of food aflatoxins using a liquid chromatograph/tandem mass spectrometer (LC/MS/MS) was conducted according to the Method of Test for Mycotoxins in Foods: Test of Multi-mycotoxin, published by the TFDA [9]. All chemicals used were of analytical reagent grade. The referral laboratory has been certified for chemical analyses of foods under the International Organization for Standardization (ISO) 17025 guidelines [16].

### 4.3. Hazard Identification and Hazard Characterization

Aflatoxins can be present in many kinds of food, including maize, tree nuts and groundnuts, etc. [11]. Acute toxicity studies of aflatoxin B1 in animals show that intake results in acute episodes of diseases including hemorrhage, liver damage, and kidney failure [17]. Aflatoxin B1 is one of the most potent known natural carcinogens [18]. Modes of action for aflatoxin B1 is digested by the p450 liver enzyme, the metabolite aflatoxin B1-exo-8, 9-epoxide binds to DNA and proteins resulting in gene mutations, which have genotoxic effects (they may damage DNA) and may cause cancer [19]. Based on epidemiological data in humans in Europe, the Joint FAO/WHO Expert Committee on Food Additives (JECFA) calculated an excess risk per unit for exposure to aflatoxin of 0.013 cancers/year per 100,000 people per ng aflatoxin/kg bw/day [20]. In other words, lifetime exposure to 1 ng aflatoxin/kg bw/day will increases the incidence of liver cancer by 13 cases per year for 100 million people. The Codex Alimentarius Commission (CAC) recommended that intake should be reduced to levels as low as reasonably possible for aflatoxin B, G, and M [21]. Many countries have set regulatory limits for total aflatoxins or aflatoxin B1, with total aflatoxins defined as the sum of aflatoxin B1, aflatoxin B2, aflatoxin G1, and aflatoxin G2 [22]. For peanuts and maize, the regulated aflatoxins limit in Taiwan is 15 ppb [23]. The sanitation standard for aflatoxins-levels tolerated in foods is shown in Table 4.

### 4.4. Exposure Analysis

The consumption and occurrence data is an important component of risk assessment. Figure 4 shows the aflatoxins hazard exposure estimates flowchart in this study. Dietary exposure to aflatoxins from peanuts and peanut products can be calculated using the following equation:

Dietary exposure = (C × CR)/BW
(1)
where BW is the average body weight of the age group, C is the concentration (μg/kg) of the mean total aflatoxin concentration in the food item according to the TFDA imported food inspections, CR is daily food consumption rate (g/kg bw/day) of the food item.

Information for CR was collected from Nutrition and Health Survey in Taiwan (NAHSIT). peanuts and peanut products daily intake data (*n* = 568) were obtained from 24-h dietary recall and correction by food frequency questionnaires. People in each subgroup were divided into five age groups: 1–2 years, 3–9 years, 10–17 years, 18–65 years, and over 65 years. Four NAHSIT surveys were conducted from 2005 to 2008 (children aged below 6 years old and above 19 years), 2010 to 2011 (junior high school students), 2011 (senior high school students), and 2012 (primary school pupils) with different target populations [24,25]. In the daily food consumption rate divided into two sub-population groups were the whole group (the population including those who eat or do not eat the peanut products) and consumer-only group (exclude the population who did not eat peanut products), these groups respectively represent the minimum and maximum possible value in daily food consumption rate of the food item. Values for BW were also derived from the NAHSIT; specific details about how to conduct surveys were presented in the Pan et al. study [25].

Aflatoxins have been classified as a group 1 carcinogen by IARC because of their adverse health consequences [4]. Due to modes of action for aflatoxins with regard to genotoxic and carcinogenic characteristics, no observed adverse effect level (NOAEL) or threshold value has been established. EFSA Scientific Committee published an opinion recommending the use of the MOE approach for the safety assessment of contaminants that are both genotoxic and carcinogenic [11]. The committee advised that risk management of genotoxic and carcinogenic compounds for public health should be based on MOE calculations using the following equation:

MOE = BMDL10/Exposure
(2)
where BMDL10 is the benchmark dose lower confidence limit of 10% of 170 ng/kg bw/day suggested by the EFSA [26].

## Figures and Tables

**Figure 1 toxins-11-00080-f001:**
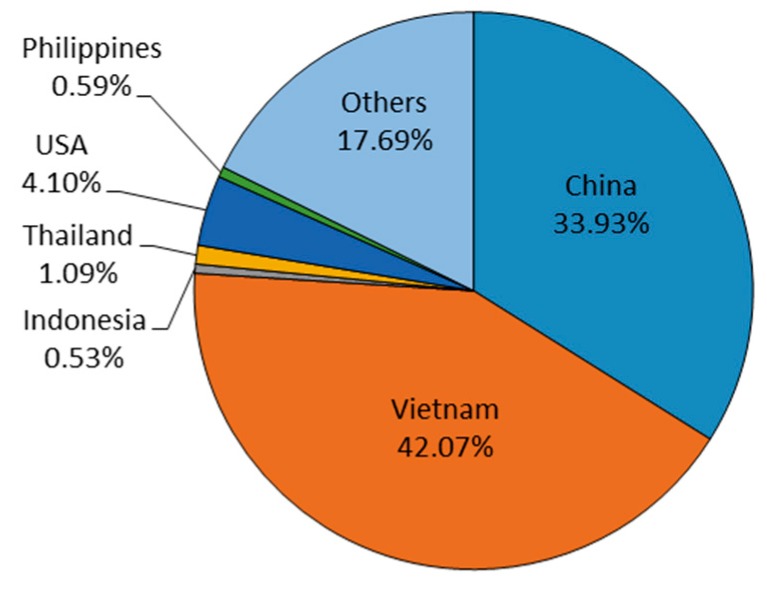
Peanut and peanut products (e.g., peanut butter, nuts, and some grain products) import weight percentage: 2011–2017.

**Figure 2 toxins-11-00080-f002:**
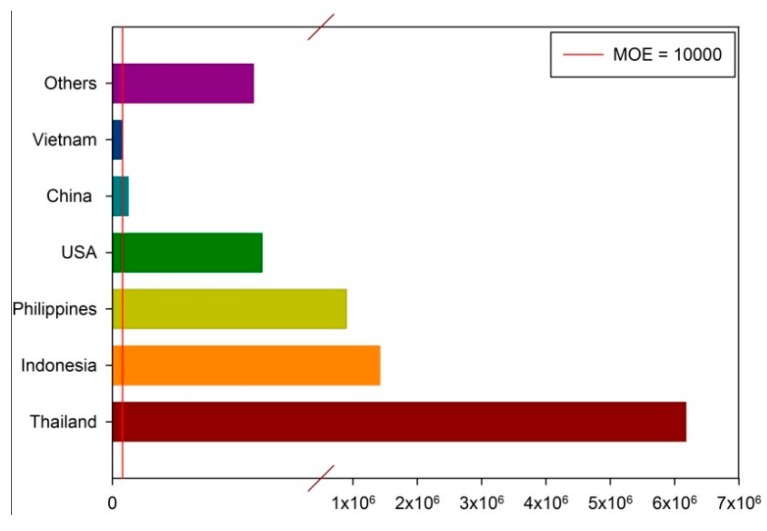
Estimation of the exposure (whole group) of Taiwanese people of all ages to aflatoxins in peanut and peanut products from various countries.

**Figure 3 toxins-11-00080-f003:**
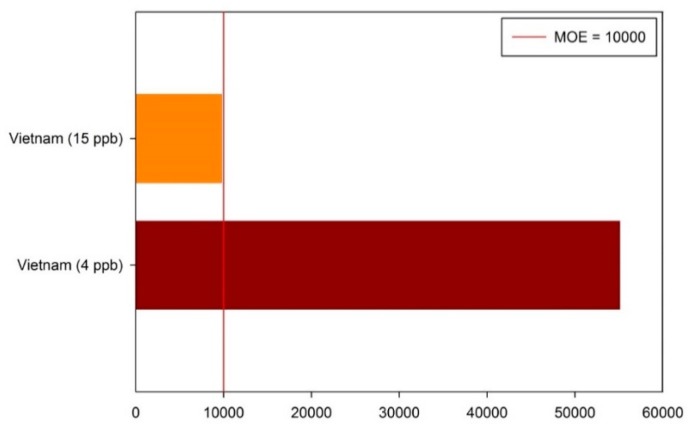
Estimation of the exposure (whole group) of Taiwanese people of all ages to aflatoxins in peanuts and peanut products from Vietnam in 4 ppb and 15 ppb maximum permitted levels of aflatoxin.

**Figure 4 toxins-11-00080-f004:**
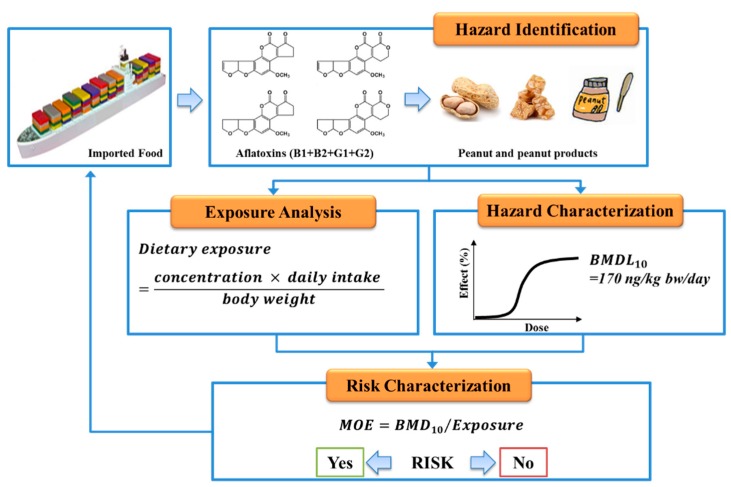
Flowchart of dietary exposure estimates for the aflatoxin in the imported peanut and peanut products in Taiwan.

**Table 1 toxins-11-00080-t001:** Mean ± standard deviation (SD) and maximum (Max) aflatoxin levels (ppb) in peanuts and peanut products samples.

Analyte	Mean ± SD	Contaminated Samples ^1^, *n* (%)	Max (ppb)	LOQ (ppb)	Exceeding Limit, *n* (%)
Aflatoxin B1	2.40 ± 21.33	262 (24%)	432.0	0.2	-
Aflatoxin B2	0.41 ± 4.41	186 (17%)	130.9	0.1	-
Aflatoxin G1	0.19 ± 3.70	37 (3.3%)	113.0	0.2	-
Aflatoxin G2	0.03 ± 0.58	20 (1.8%)	17.0	0.1	-
Aflatoxins ^2^	3.03 ± 23.71	267 (25%)	441.0	-	34 (3.1%)

^1^ Number of samples: 1089, ^2^ aflatoxin (B1 + B2 + G1 + G2).

**Table 2 toxins-11-00080-t002:** Mean ± standard deviation (SD) and maximum (Max) aflatoxin levels (ppb) in peanuts and peanut products samples from different countries.

Countries	No. of Samples	Mean ± SD	Contaminated Samples, *n* (%)	Max (ppb)	Exceeding Limit, *n* (%)	Mean ^1^
China	49	0.947 ± 1.953	17 (35%)	8.1	0 (0%)	0.947
Vietnam	307	4.180 ± 24.555	75 (24%)	258.3	13 (4%)	1.206
Indonesia	80	11.805 ± 50.257	24 (30%)	412.0	10 (13%)	0.660
Thailand	103	1.562 ± 14.111	13 (13%)	143.0	2 (2%)	0.074
USA	97	1.372 ± 3.892	41 (42%)	28.0	2 (2%)	0.843
Philippines	109	1.053 ± 2.363	51 (47%)	14.2	1 (1%)	0.931
Others	344	1.784 ± 24.002	46 (13%)	441.0	4 (1%)	0.207

^1^ Excluding the samples excess of maximum aflatoxin levels.

**Table 3 toxins-11-00080-t003:** Estimation of the MOE levels of aflatoxin in peanut and peanut products from different nations.

Country	Scenario	Group				
		1–2	3–9	10–17	18–65	above 65
China	W ^1^	11,711	12,528	30,136	10,581	17,634
	C ^2^	371	830	1710	1047	1368
Vietnam	W	7417	7934	19,087	6701	11,169
	C	235	526	1083	663	866
Indonesia	W	1,074,443	1,149,382	2,764,939	970,750	1,617,917
	C	34,032	76,164	156,870	96,044	125,476
Thailand	W	4,680,276	5,006,707	12,044,076	4,228,589	7,047,649
	C	148,244	331,770	683,325	418,369	546,574
USA	W	108,910	116,506	280,266	98,399	163,999
	C	3450	7720	15,901	9735	12,719
Philippines	W	680,404	727,859	1,750,929	614,739	1,024,565
	C	21,551	48,232	99,340	60,821	79,459
Others	W	102,786	109,955	264,507	92,867	154,778
	C	3256	7286	15,007	9188	12,004

^1^ W: whole group; ^2^ C: consumer only.

**Table 4 toxins-11-00080-t004:** Sanitation standards of aflatoxin levels tolerated in foods in Taiwan.

Food Category	Tolerance of Total Aflatoxin (Including Aflatoxin B1, B2, G1, G2)
Peanut, corn	Not more than 15 ppb
Rice, sorghum, legumes, nuts, wheat barley and oat	Not more than 10 ppb
Edible oils and fats	Not more than 10 ppb
Milk	Not more than 0.5 ppb (as aflatoxin M1)
Milk powder	Not more than 5.0 ppb (as aflatoxin M1)
Other foods	Not more than 10 ppb

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
