# Peer review of "Assessing Aflatoxin Exposure Risk from Peanuts and Peanut Products Imported to Taiwan"

_toxins, 2019, doi:10.3390/toxins11020080_

Round 1
Reviewer 1 Report
The article is a complete study in order to assess Taiwan population exposure to aflatoxin from peanuts and peanut products in the period from 2011 to 2017, in fact a total of 1089 samples have been collected and analyzed. The important aspect is to know how has been collected the information for Consumption Rate, which was from Nutrition and Health Survey in Taiwan (NAHSIT) and in different years. So that, please justify why it was used that option to estimate the consumption of different products of peanuts during 6 different years.
In lines 99 to 103, it is cited a old reference with information from 1982 to 1994, just to compare the results obtained in the present study. Due to its a oldest information please delete this and check in all the article possibles citations like this.

Author Response
Q1: Its not clear whether or not samples and results presented in Table 2 are from a different monitoring study from those presented in Table 1.
Ans: Table 1 and 2 are from the same raw data. We depict this in line 69. Table 2 concentrations are from a monitoring study from those presented in Table 1.
Q2: In abstract at lines 6 and 7 authors mention a 1089 samples of peanuts products although peanuts are mentioned in line 6. In Results and scission section lines 50-51 still 1089 samples are only from peanut products . At lines 68-69 authors mentioned "In this survey...." which according to my understanding they presented a different study from the one of 1089 samples and in this peanuts and peanuts products different from the previous one are mentioned (only peanut butter is in common). At lines 103-106 there are mentioned again both peanuts and peanuts products. I believe that authors should clarify this crucial point in order to facilitate review procedure.
Ans: Our samples including peanuts and peanut products, such as roast peanuts, garlic flavor peanuts, peanut candy, peanut ice cream roll, peanut butter, and peanut flour and so on. Line 6-7, 50-51, and 109, we revise that depiction in the text to peanuts and peanut products.

Reviewer 2 Report
Its not clear whether or not samples and results presented in Table 2 are from a different monitoring study from those presented in Table 1.
In abstract at lines 6 and 7 authors mention a 1089 samples of peanuts products although peanuts are mentioned in line 6. In Results and scission section lines 50-51 still 1089 samples are only from peanut products . At lines 68-69 authors mentioned "In this survey...." which according to my understanding they presented a different study from the one of 1089 samples and in this peanuts and peanuts products different from the previous one are mentioned (only peanut butter is in common). At lines 103-106 there are mentioned again both peanuts and peanuts products.
I believe that authors should clarify this crucial point in order to facilitate review procedure.
Author Response
Q1: The article is a complete study in order to assess Taiwan population exposure to aflatoxin from peanuts and peanut products in the period from 2011 to 2017, in fact a total of 1089 samples have been collected and analyzed. The important aspect is to know how has been collected the information for Consumption Rate, which was from Nutrition and Health Survey in Taiwan (NAHSIT) and in different years. So that, please justify why it was used that option to estimate the consumption of different products of peanuts during 6 different years.
Ans: This is one of the research Limitations. Chinese foods are complex and dynamic, and it is very difficult to match exactly the food groups for items to be surveyed with the foods actually consumed. Taiwan's peanut and peanut products mainly rely on imports. According to the statistics of import and export agricultural data, the total import weight of peanut and its products in 2008 was about 14,934 metric tons. In 2017, the import weight of peanut and its products was about 15,626 metric tons, although there is a growing trend (about 5%), the change is not significant. So the consumption rate conducted from 2005 to 2008 (children aged below 6 years old and above 19 years) can be used before the new food intake survey data release is available.
Q2: In lines 99 to 103, it is cited a old reference with information from 1982 to 1994, just to compare the results obtained in the present study. Due to its a oldest information please delete this and check in all the article possibles citations like this.
Ans: We already deleted line104 to 108 and we keep line30 to 32 due to this reference summarizes the aflatoxin occurrence in food from different countries.
